# Tacit Contributions and Roles of Senior Researchers: Experiences of a Multinational Company

**Walter Pérez Villa [1], Amaya Pérez-Ezcurdia [2] and Miguel Angel Vigil Berrocal [1,*]**

1. Department of Mines Exploitation and Prospecting, Project Engineering Area, University of Oviedo, 33004 Oviedo, Spain
2. Department of Engineering, Public University of Navarra, 31006 Navarra, Spain
* Correspondence: vigilmiguel@uniovi.es

**Abstract:** One of the concerns of innovation-dependent organisations is that the gradual increase in the average age of their employees might affect their creativity and innovation rates, leading to losses in competitiveness. The purpose of this paper was to deepen the identification and understanding of the contributions done by senior researchers within a private organisation. This study was based on field qualitative research on a multinational company. Interviews were performed were senior researchers and the transcripts were analysed with a qualitative data analysis (QDA) software to organise, analyse and find insights in unstructured or qualitative data. Analysis was performed using axial coding, which relates data together to reveal codes and categories from participants' voices within the collected data. The points of view of senior researchers were explicitly sought and the findings indicated that these veteran professionals can be more valuable for their contributions as experienced workers than for their scientific productivity at the individual level, without disregarding it. Senior researchers have acquired tacit skills linked to their experience, such as a holistic view of the issues and efficient work methodologies. Therefore, they develop formal or informal roles over time related to advice and knowledge transfer. Consequently, it was found that their tacit contributions and roles increase the intellectual capital of the organisation. This paper helps in understanding the contributions made by senior researchers within a private organisation. No other reviews have sought to obtain such information on this specific sector.

**Keywords:** tacit contributions; intangible assets; senior researcher perspective; senior researcher role; senior researcher motivation; ageism; age management; patenting; industrial researchers; intellectual capital; knowledge transfer

## 1. Introduction

An obvious problem faced by most developed economies is the ageing of its population, particularly its active workers (Ogura and Jakovljević 2014). This study focused on the contributions and roles that senior researchers make in the field of research and innovation, considering the latter as a key factor in the competitiveness and survival of an organization (Ortt and van der Duin 2008).

It has been considered that, in general, the relation between age and creative productivity of people has an inverted beta distribution curve shape, with a sharp increase in the first years and a decline in the final years of the person (Simonton 1997). However, more and more voices are calling into question that idea, at least in part. For instance, Appelbaum et al. (2016) identified a number of old-age stereotypes that have decreased organisational productivity in empirical studies, including reluctance to change, decreased learning ability, intelligence and memory, poor health and accidents, higher organisational costs, decreased motivation, and low innovation and productivity. Rietzschel et al. (2016) affirmed that whether or not certain employees (either young or old) might show the same level of creativity and innovative behaviour as others in their age, such features will depend

not only on their personal attributes but also on the context they are working in. At the same time, whether or not certain contexts might be more conducive to creativity or innovation, the results will depend on the personal characteristics (including age) of the people working within those circumstances. On the other hand, Frosch (2011) considers that there exist doubts with respect to the methodology used in several studies reaching the previous conclusion. Frosch's hypothesis is that most results on such studies are biased in favour of the youngest workers; in particular, most articles do not take into account the possible contributions of the seniors in other aspects more difficult to measure, termed by the author as tacit contributions. Tacit contributions emanate from the concept of tacit knowledge that McAdam et al. (2007) define as "*knowledge-in-practice developed from direct experience and action; highly pragmatic and situation specific; subconsciously understood and applied; difficult to articulate; usually shared through interactive conversation and shared experience.*"

The above information brings us to our research question:

*RQ:* **What are the tacit contributions that older researchers provide within an organisation, contributing to its inventive productivity, to a greater or lesser extent?**

This is an issue that has not been sufficiently addressed, according to the limited literature available on the subject. In fact, there are hardly any studies that address this problem from the point of view of senior researchers themselves. The results of this investigation provide more knowledge on the contribution of the most senior scientific workers on the innovative capacity of an organisation. Not including senior scientists, authors such as Jotabá et al. (2022) proposed a literature review on human resources developments through the adoption of innovative practices in similar organizations, including a section on motivations and obstacles to the adoption of such practices; they considered some characteristics that could be relevant on the research proposed by this article, however they suggest it as a general knowledge and learning scheme in the framework for adopting innovation in human resources management.

A qualitative research study was conducted in a multinational company in the steel sector with personal interviews of a representative group of senior researchers that addressed a set of questions.

The remainder of this paper is structured as follows: Section 2 contains the conceptual framework. The methodological details of the research are explained in Section 3. Section 4 gives the results of the case study, as well as the discussion of them. Finally, Section 5 summarises the main conclusions, limitations, and future research.

## 2. Literature Review

### 2.1. Age and Individual Productivity

There is a great need for a better understanding of the associations of age with creativity and innovation. Relations between age and creativity at work are still largely understudied. One exception is the professional domain of science, which illustrates the complexity of that relationship (Rietzschel et al. 2016).

As stated by Simonton (1997), and cited by hundreds of studies, an inverted beta distribution curve shape relates the age and productivity of career trajectories. However, significant differences have been found depending on the scientific field. Rorstad and Aksnes (2015) and Sugimoto et al. (2016) clearly distinguish between the social sciences, where productivity does not clearly decrease with age, and the technical disciplines. Within the technical disciplines, differentiated curve profiles have also been found (Gonzalez-Brambila and Veloso 2007; Costas et al. 2010; Rorstad and Aksnes 2015). Goodwin and Sauer (1995) differentiate between highly and poorly productive profiles, finding that the decline is much smaller among the most productive researchers.

Sturman (2003) performed a meta-analysis on the subject and concluded that there is no inverted U shaped (or beta shaped) relationship between age and performance in all time variants and work contexts. Therefore, the prediction of performance over time depends on the characteristics of the job complexity and the performance measurement system. The study of Rorstad and Aksnes (2015), involving almost 12,400 Norwegian university

researchers, shows that academic position is more relevant for academic productivity than age and gender. In the analysed fields, a regression model showed that researchers' age can only explain 13.5–19% of the variance in the publication output at the levels of individuals, meaning also that most of the variance in publication rates is due to other factors.

Similar conclusions are reached in the private sector: Rietzschel et al. (2016) state that empirical literature does not support direct or zero-order relations between age and productivity, and that more complex relationships (moderated by contextual or other individual factors) are more plausible, indirect, and curvilinear. Some of the influencing factors have been analysed. For instance, among others, the influence of the type of career that the professionals had chosen (Tremblay et al. 2002; Manolopoulos et al. 2011), the influence of a company change (Fallah et al. 2012), the motivations of the inventors to patent (Mathew and Chakraborty 2012; Blind et al. 2022), the size of the organisation (Schettino et al. 2013) and personal traits (Zwick et al. 2017). Regardless of the above factors, the number of patents held by researchers is not correlated with the average value of those inventions (Mariani and Romanelli 2007), not to mention the informality of much innovative activity (Hall et al. 2014), and the extensive efforts of the scientific community to find robust complementary innovation metrics. Many other studies linking productivity decreases with age, use the number of filled patents as indicators; however, the latest findings challenge its suitability. Consequently, the next hypothesis can be formulated:

**H1:** *The number of patents of an industrial researcher is not a good enough indicator of his/her productivity over time.*

### 2.2. Age and Firm-Level Productivity

As far as innovation is concerned, the literature survey does not find conclusive evidence stating that a youth-centred human resource strategy (always) fosters innovation (Frosch 2011). Park and Kim (2015) concluded that workforce ageing would have a positive influence on exploitative innovation performance and had an inverted U-shaped relationship with exploratory innovation performance. In addition, age diversity only attenuated the positive workforce ageing and exploitative innovation performance relationship. Recently, Sung and Choi (2021) found in Korea that age diversity in high-tech firms, with a relatively young workforce, increased firm innovation; additionally, they observed that age diversity indeed increased firm innovation for high-tech firms, but not for non-high-tech firms; consequently, only high-tech firms could increase innovation by enhancing age variations among employees, considering their relatively young and age-homogeneous workforce. On the other hand, Mothe and Nguyen-Thi (2021) found that the effect of age diversity on innovation depends on the age distribution pattern of employees: positive for firms characterised by heterogeneous age groups (variety), negative for those dominated by polarised age groups (polarisation).

The tenure of employees, an age-related aspect in some ways, has also generated interest. For instance, according to Chen et al. (2012), tenured employees were found to be more committed to organisations; however, they did not find a significantly positive effect of personal age on commitment. Similarly, longer firm-specific and industry tenure of employees would enhance the positive effect of firm age on the quality of exploitative innovations, while amplifying the negative effect of firm age on the quality of explorative innovations (Tschang and Ertug 2016). So, the negative effects of firm age on the quality of explorative innovations could be mitigated, as used by the firm, band y talent resources (employees) who have lower firm-specific and industry-wide tenure.

Organisations deploy their human resources policies according to various objectives: one of them may be to balance the age distribution of employees, and another fundamental one is to acquire and retain knowledge; and the hiring of successful inventors is a classic method to achieve the goal. Knowledge retention can be considered as a matter of turning the individual explicit or tacit knowledge, as well as the knowledge of older employees, into organisational knowledge. Such knowledge might be formalised and codified explicitly enough into ways of work that can become standardised, in order to guide younger

employees in their work. In this context, tacit term refers to something that is understood without being expressed directly (Wikström et al. 2018). According to Polanyi (1958), tacit knowledge has a personal quality, which makes it hard to formalise and communicate: it is deeply rooted in action, commitment, and involvement in a specific context.

The dimension of tacit knowledge is crucial in the current environment of rapid cycle time, short product lifespans, and increasing emphasis on exploratory innovation (Lee et al. 2016). The review of Thomas and Gupta (2022) found that tacit knowledge sharing is a key behaviour in innovative organisations.

Both scientific and technological inputs to innovation embody a considerable tacit component which can only be acquired by practical experience. Indeed, tacit knowledge and skills are particularly significant to scientific methodology and the scientific view of the world (Senker 2008). So, the next hypothesis can be formulated:

**H2:** *The productivity of an industrial researcher does not necessarily increase or decrease over time, but changes its nature, as he or she acquires tacit skills linked to experience.*

### 2.3. Tacit Contributions of Senior Inventors

Nonaka (1994) proposed a paradigm for managing the dynamic aspects of organisational knowledge creating processes. Its central theme is that organisational knowledge is created through a continuous conversion between tacit and explicit knowledge. For Nonaka (1994), there are four different modes of knowledge conversion: (1) from tacit knowledge to tacit knowledge through socialisation; (2) from explicit knowledge to explicit knowledge through combination; (3) from tacit knowledge to explicit knowledge through externalisation; and (4) from explicit knowledge to tacit knowledge through internalisation.

A similar adjective to tacit, and perhaps more used in management literature, is intangible. This can be intangible knowledge or, from a broader point of view, intangible assets. Most definitions do converge on the "immaterial" aspect of these assets: they have neither physical substance nor specific monetary value, yet they significantly contribute to value creation for a business (St-Pierre and Audet 2011). Intangible is something that is impossible to touch, to describe exactly, or to give an exact value. Although tacit and intangible are associated with different biological senses, they both have traditionally been used to describe knowledge, because sometimes knowledge is difficult to describe, to formalise and to transfer. However, Kristandl and Bontis (2007) affirm that it is possible to find and propose a common definition for intangibles, derived from the resource-based view: according to them, intangibles are strategic firm resources that enable an organisation to create sustainable value, but are not available to a large number of firms (rarity); They lead to potential future benefits, which cannot be taken by others (appropriability), and are not imitable by competitors, or substitutable using other resources; They are not tradeable or transferable on factor markets (immobility) due to corporate control; Because of their intangible nature, they are non-physical, non-financial, are not included in financial statements, and have a finite life. In order to become an intangible asset included in financial statements, these resources need to be clearly linked to a company's products and services, identifiable from other resources, and become a traceable result of past transactions. Items such as image, reputation, information technologies, customer portfolio, flexibility, knowledge domain, skilled employees, brands, patents, among others, are indispensable in the organisational environment (Osinski et al. 2017). It is argued that organisations can substitute tangible assets and resources, but they are unlikely to do that with intangible assets (Yaseen et al. 2016).

Another similar denomination is intellectual capital. Intellectual capital (IC) is a set of intangible assets that the firm owns or has access to (Edvinsson and Malone 1997): it has been related to innovation (Buenechea-Elberdin 2017) or to performance (Pedro et al. 2018). Chatterjee et al. (2022) studied the moderating effects of age in the impact of intellectual capital on firm performance: they found that the effect of young adults is greater than that of middle-aged adults.

The most frequently used groups of components, in studies dealing with intellectual capital's influence on performance, correspond to a triad of human capital, structural (organisational or process) capital, and relational (social or customer) capital. In short, human capital refers to the (tacit) knowledge and skills possessed by the people of an organisation, structural capital encompasses the organisation systems of codified knowledge (databases, processes, patents, etc.), and relational capital refers to an organisation's external networks and interactions. They all determine positively the performance of organisations/regions/countries, but their influence is not linear and depends on various factors associated with the context and surrounding environment (Pedro et al. 2018).

Wang et al. (2019) add a component to the three traditional ones: the psychological component. In fact, they apply the concept of intellectual capital to entrepreneurship, and replace structural capital with psychological capital. They propose the concept of entrepreneurial intellectual capital, which consists of human (i.e., age and education, graduate work experiences, non-graduate work experiences, role models), relational (i.e., trustworthiness and co-founder relations) capitals, and psychological capital (optimism, self-efficacy, hope, and strength).

The age of the company appears many times as an influencing factor in intellectual capital, but the age of the employees has rarely been considered. According to Yaseen et al. (2016), the effect of the relational capital on competitive advantage is moderated by age, and it is stronger among younger men. Ginesti (2019) analysed CEO's features and intellectual capital and found evidence that companies with older CEOs demonstrate better intellectual capital efficiency.

Tacit knowledge will continue to play an imperative role in innovation, most significantly due to the complexity of systems and the emergence of new technologies (Senker 2008). It is interesting to know how older researchers can contribute to the improvement of some of the intangible assets of an organisation that is committed to innovation. In view of the above, the following hypothesis is proposed for the research:

**H3:** *Tacit knowledge will most likely be the most obvious tacit contribution of senior researchers, enriching the intellectual capital of the organisation.*

*2.4. Role of Senior Researchers*

There exist few studies that address the roles that senior workers play in their organisations. Almost all of them are confined to health or school environments. In any case, the human resources literature recognizes the role of veteran workers as repositories of organisational memories and potential mentors (Dunham and Burt 2011). Anyway, besides their age, the contributions-roles are due to experience and, in particular, to time in the organisation (Dunham and Burt 2011).

Typical mentoring programs pair experienced employees (mentors) with younger employees who have less experience (protégés or mentees) within a relation of 6 to 12 months (Single and Muller 2001). Hopkins-Thompson (2000) distinguishes between mentoring and coaching. The mentor is more focused on the mentee socialisation process in the organisation, while the coach focuses more on the specific skills necessary to perform a certain job. Abbidin (2006), on the other hand, focuses on the difference in the methods to be used, since the coach instructs, while the mentor advises. In this paper, mentoring and coaching will be considered as synonyms.

According to Kram (1983), mentoring functions at the workplace can be classified in two different categories: the first one is related to career functions, which are the aspects of the relations that primarily enhance career advancement, such as sponsorship, exposure-and-visibility, coaching, protection and challenging assignments; The second group of functions are related to the psychosocial aspect, which enhance the sense of competence, clarity of identity and effectiveness in the managerial role; those functions are role modelling, acceptance-and-confirmation, counselling and friendship.

The benefits are not only for the mentee. There are many benefits for the mentor, including the personal satisfaction of observing and participating in the success of their

mentees, improvement in their job performance by providing them with new perspectives and knowledge, and learning new skills such as those related to emerging technologies from their protégés (Parise and Forret 2008). Additionally, mentors may gain recognition among peers and superiors for helping to develop high-potential individuals within the organisation, as well as experimentation of feelings of generativity or immortality from watching their mentees succeed.

Organisations can also obtain notable benefits from mentoring practices (Short 2014): (a) Enhanced leadership capability; (b) Knowledge transfer; (c) Role modelling/credibility; d) Access to experience; (e) Improve communications; (f) Employee retention/engagement. Swap et al. (2001) concluded that mentoring and storytelling can leverage the knowledge of an organisation, particularly its tacit knowledge, to build core capabilities. Face-to-face interaction is the primary means for tacit knowledge sharing (Wang and Wang 2012). Mentoring can be seen as a micro-level knowledge-producing community of practice. Mentors transfer tacit knowledge to both mentees and organisations (Singh et al. 2002). Bryant (2005) found that higher perceived levels of peer mentoring were related to higher perceived levels of knowledge creation and sharing. Definitely, coaching and mentoring are seen as competitive drivers to cultivate innovation and creativity in turbulent business environments (Woo 2017).

There are also no significant studies that address the role or roles that senior workers can play in research organisations. Cohen et al. (2012) suggest that mentoring programs can enhance research productivity while incorporating accountability features like formalised reports of progress and mentorship feedback. Al-Zoubi et al. (2019) found that mentoring had a positive effect on the creation of new innovative ideas. The following hypothesis can therefore be envisaged:

**H4:** *Mentoring-coaching is the main role senior researchers play or could play in research organisations.*

## 3. Materials and Methods

To answer the research questions, an exploratory and explanatory research throughout a single case study was conducted, as it is the most suitable methodology for this type of research (Eisenhardt and Graebner 2007; Yin 2017). This case study was done at one of the largest steel manufacturing companies (>150,000 employees worldwide), with research teams located at eleven facilities, in seven countries. This case was chosen because it is a company with a long history and experience which is based worldwide and is part of a very mature and developed sector. This company, as expected, has employees of all ages, including the R&D staff. The terms veteran, senior, senior researcher and senior inventor are used indistinctly in this study. It has been determined by the authors that a senior researcher is a person with a technical background (engineering or sciences studies) and more than 15 years of experience in research. To enrich the answers of the study and avoid the saturation of feedback, the authors invited also some researchers, close to the proposed age range, that were well compromised on research, as confirmed by other interviewees.

The company of the study, as most of the industrial conglomerates, has been formed by merging smaller companies. At least fifty companies, along with a century of existence, have merged to get to the point where this enterprise is today. The business mergers, besides the evident improvements on organisational standards and production capabilities, fomented the encounter of several work cultures, and, as seen with the different R&D centres, a research network has allowed the specialisation and the capabilities and synergies to allow innovation at both the product and process levels in the organisation.

In the first place, the scientific productivity of the researchers measured as the number of granted patents has been analysed. The European Patent Office (EPO) has been consulted as a principal source of information, which has provided the data related to the surveyed enterprise. EPO does not only consider the patents granted in the European Union, but also those conceded world-wide. Concerning the company mergers, information was sought about the laboratories and researchers associated with the original companies, those

preceding the mergers. It is important to remark that the inputs of those mother companies were considered in the study.

In the study, a quantitative analysis was done concerning the patents granted, in the last 30 years, to the researchers of the analysed company and those who were part of the mergers. With the information found on social networks and websites specialising in papers and patenting publications, the age range at the moment of each granted patent was determined. In total, a database of at least 450 patenters was obtained: all of them are or were part of the studied company or the merged ones. Among other results, the described analysis allowed the identification of key people in the company from a scientific productivity point of view. All the interviewees, except one person working at the patent department, have several granted patents. All of them are still associated with research, but in some cases that link comes as a manager of the research itself. In spite of the diverse origin of the researchers, most of the interviewee answers were very uniform, causing saturation, forcing the authors to limit the number of interviewees to 10, following the considerations of Baker and Edwards (2012). Table 1 summarises the main characteristics of the interviewed professionals.

**Table 1.** Summary of the professionals interviewed in the study.

| Interview Number | Age | Sex | Professional Experience | Career | Scientific Productivity |
|---|---|---|---|---|---|
| 1 | 46–50 | M | 21–25 | Researcher-R&D manager | Decreasing |
| 2 | 51–55 | F | 21–25 | Researcher-R&D manager | Decreasing |
| 3 | 46–50 | M | 21–25 | Researcher-R&D manager | Decreasing |
| 4 | 36–40 | M | 5–10 | Researcher | Decreasing |
| 5 | 51–55 | M | 31–35 | Researcher-R&D manager | Decreasing |
| 6 | 51–55 | F | 26–30 | R&D manager | - |
| 7 | 41–45 | M | 16–20 | Professional-Researcher | Constant |
| 8 | 36–40 | M | 11–15 | Researcher | Concentrated |
| 9 | 41–45 | M | 11–15 | Researcher | Concentrated |
| 10 | 41–45 | M | 21–25 | Professional-Researcher | Concentrated |
| 11 | 41–45 | M | 16–20 | Researcher | Concentrated |

Following Miles et al. (2020), the qualitative data analysis entails three concurrent flows of activity: data condensation, data display, and conclusion drawing/verification.

All interviews were done face to face or via video call by the same researcher and the average length was one hour. After asking permission of the interviewees, each interview was recorded in order to be transcribed afterwards. Concerning the content of the interviews, the first part was about asking the personal biography of the senior researcher, emphasising the motivations they have for applying for patents and their own professional life choices and decision-taking. For this part of the interview, a scholarly chronicle style was adopted, one of the most fundamental types of biographical research (Eaton 1964), which focuses on the historical path of the person, telling his/her story in chronological order with an emphasis upon developments of particular plots on his/her scientific development, including detailed descriptions of particular acts of recognition or notoriety. The second part of the interview focused on the contributions that a senior researcher, by the fact of being one, makes to his/her research team and, therefore, to the company where he/she works. In this case, it was not his/her personal contributions, but those of any professional dedicated to research: the purpose was to collect the vision of the interviewees, forged after collaborating for many years with several professionals, both in- and outside their organisation.

Transcripts were analysed with Nvivo® v11 software. NVivo® is a qualitative data analysis (QDA) computer software package produced by QSR International that allows

to organise, analyse and find insights in unstructured or qualitative data like interviews, open-ended survey responses, journal articles, social media and web content (McNiff 2016).

The analysis was performed using the research technique known as axial coding (Merriam and Grenier 2002), that involves relating data together seeking to reveal codes, categories and subcategories grounded within participants' voices within the collected data.

The interviews were designed following an open-answer questionnaire that allowed the interviewees to express themselves freely. An initial thematic-type code structure was built through the compilation of passages of text linked by a common theme or idea. This allowed the authors to index the text into categories derived from the blocks of questions used in the interviews. Afterwards, a coding of all the interviews was carried out using this arrangement and, during the analysis process, the structure was slightly readjusted. In addition, following the recommendations of Saldaña (2021), several memos were written with the ideas and relationships that seemed to emerge from the material obtained in the study.

Finally, a second round of relational reading of the transcripts was done, and a conceptual coding was created. In this case, the reading was more analytical, looking over the specific ideas and searching for patterns among the different respondents. Likewise, relationships between codes and interviewees were sought, based on their biographical characteristics. At the end, a definitive selective coding was performed and, as a result, the emerging categories constituting the main conclusions of the study were established. All three rounds of coding were performed by one of the authors, but each one was reached by the consensus of the three researchers of the study. Figure 1 represents the different phases of the study.

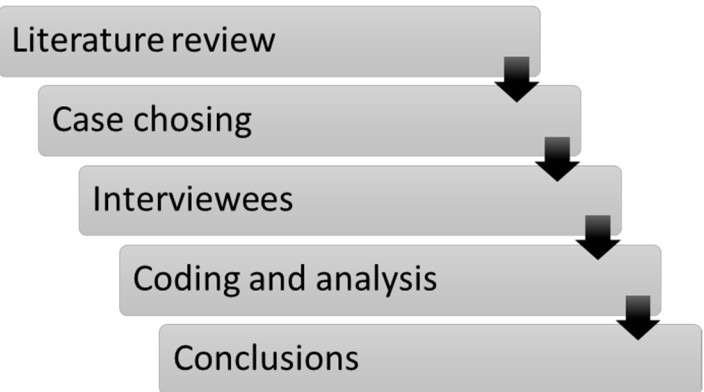

**Figure 1.** Phases of the study.

## 4. Results and Discussion

The outcomes of the interviews were condensed in a hierarchy chart known as "*tree map*", shown in the Figure 2. This chart represents particular words or concepts by a rectangle whose surface is directly proportional to the times they appear on the interviews. In order to elaborate the graph, words referring to names or irrelevant terms (for instance, "also", "then", etc.) were eliminated. Next, the first 30 concepts were taken into account. These concepts group similar terms, as in "research" that includes the words "research", "researches" and "researchers".

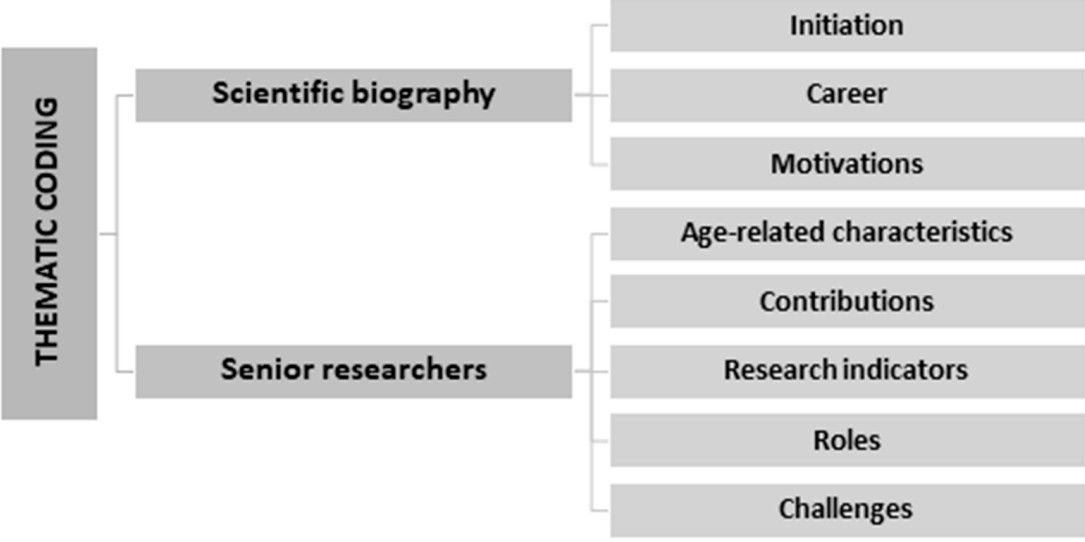

**Figure 2.** Tree map of interviews.

It can be seen that the most frequently mentioned words during the interviews were "research" and "patents", followed by other related ones such as "think", "knowledge", "people", "industry", "senior", and "engineers", easily explained by the nature of this study. Moreover, related concepts also appeared very frequently, such as "motivation", "experience" or "ideas". It is also worth noting the appearance of "young" as opposed to "senior" and the lack of preeminence of the word "publications", highlighted their limited importance in the industrial domain.

The code structure was used afterwards: Figure 3 presents the thematic coding and Figure 4 exposes the axial coding.

**Figure 3.** Thematic coding of interviews.

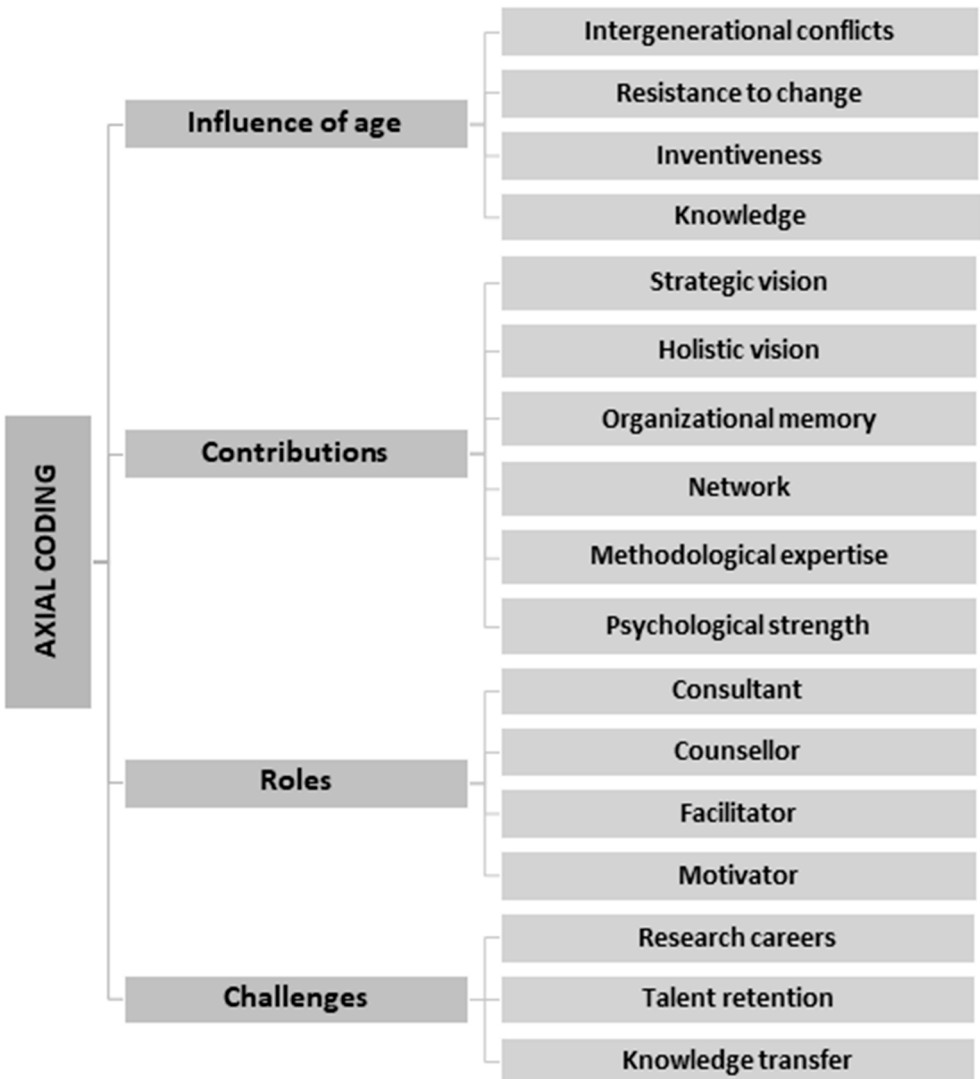

**Figure 4.** Axial coding of interviews.

Subsequently, the results obtained are shown and discussed based on three axes: Scientific productivity, Tacit contributions, and Roles performed.

### 4.1. Scientific Productivity along the Professional Career

All interviewees were considered to be great experts on their field by their research colleagues. However, none of them stated that scientific productivity had a clear rising trend over time when the scientific productivity was measured as the number of patents per period of time. In this sense, a group of interviewees clearly recognized that their productivity had decreased because they had moved towards managerial positions.

Among the interviewees that continued working on research after several years, only one of them clearly recognized a decrease in their productivity and who could not find an evident motivation to patent. For the rest of the active researchers, it was not possible to envisage any trend, since productivity was concentrated for them in certain years, coinciding with their participation in specific projects that, due to their characteristics, had given place to patent applications.

In general terms, it is stated that patenting is not an objective by itself, but rather a part of the job; and that the decisions to apply for a patent are generally taken by the organisation at higher levels instead of by the researchers or team leaders. In spite of that, it was understood that once the patent was granted, the researchers felt motivated because they realized that the organisation had appraised their work.

"For researchers to be productive in terms of patents and articles, first, the company has to decide if that is the most important thing, or if it is more important what might be accomplished within the company while, at the same time, it could be useful for the business." (Interview number 8).

Table 2 presents a list of the motivations that might have a company for patenting, according to the opinion of the interviewees.

**Table 2.** Motivations to patent in a mature industrial company.

| Personal Motivations | Organisational Motivations |
|---|---|
| • Economic recognition<br>• Personal recognition<br>• Freedom to research, collaborate, etc. | • Market recognition<br>• Protection of products and services developed by the company<br>• Protection of ideas susceptibles for development<br>• Know-how protection before starting an R&D collaboration |

Clearly supporting the hypothesis no. 1 (H1), there was unanimity among all interviewees when considering the number of patents as an insufficient indicator to measure the individual productivity of an industrial researcher. Likewise, all of them recognized the difficulty of defining good indicators because, for example, a time perspective is needed when evaluating the impact that novel ideas have had either on the organisation or the market. Some indicators mentioned by the interviewees included: the number of projects being worked on, number of launched ideas, participation at different research forums, peer recognition, and support to young researchers in the organisation (mentoring).

A basic objective of this research work was to reveal the opinion that senior inventors have about the influence of age on different aspects of professional researchers' performance. It is worth mentioning that none of the interviewees considered that creativity diminishes over time, but rather that it is a matter of personality. On the other hand, concerning other capacities such as drive or energy at work, change resistance, or generation of disruptive ideas; there was no unanimity, although there was a majority position that establishes that age is not determinant on such parameters.

### 4.2. Tacit Contributions of Senior Researchers

When asked about the contributions that senior researchers can provide to their team and their organisation, almost all the interviewees agreed on the explicit knowledge that they hold: a senior is usually considered an expert.

"[Senior researchers] have a lot of knowledge and work with so many subjects."

(Interview number 4)

"[Senior researchers] can avoid reinventing the wheel: that's also something I've faced with [young] people coming with a good idea, but such an idea has been already known for 20 years in another domain or in another lab, and they are reinventing the wheel. With more experienced people, they know what the real level of prior art is, not only inside the organisation, but outside as well."

(Interview number 6)

Furthermore, the interviewees commented that older researchers do not only possess a good amount of knowledge about their area of expertise, but they also understand other fundamental issues that might be included in what has been called intangible or tacit knowledge. For instance, respondents were unanimous on the idea that seniors have a strategic and holistic vision of their area of study and the organisation market.

The second most commented aspect is that seniors have organisational memory: they have seen the results of several projects, including some that were successful and some others that did not come to fruition. Seniors comprehend the reasons for many of the

decisions that have been made over time, they know the resources of the organisation, and can anticipate the problems that may arise in a given project.

Half of the interviewees highlighted the abilities that most veteran researchers have in terms of research methodology, including the knowledge related to the proper way to expose the results when applying for a patent. The ability to plan and organise a research project was also highly valued, as was the mastery of the most appropriate techniques and the selection of the most interesting partners in research.

Finally, it is worth noting the importance that interviewees give to the network of contacts that a senior can offer, both within and outside the organisation. Senior researchers have had the chance to meet several people in events or have had multiple partners; they are able to offer such contacts for the benefit of the research.

Table 3 offers a compilation of the competencies, including tacit contributions, that were identified by the interviewees as the most relevant in industrial senior researchers.

**Table 3.** Specific competencies of senior researchers in an industrial organisation, as identified by the interviewees.

| Specific Competencies of Senior Researchers in an Industrial Organisation | |
|---|---|
| • Identification of missing knowledge<br>• Mastery of knowledge sources<br>• Identification of R&D trends in the medium and long term<br>• Identification of industry trends<br>• Identification of key knowledge for the company<br>• Ability to identify whether the ideas to be developed can have industrial applicability<br>• Provision of context to knowledge (senior researchers are able to see the bigger picture)<br>• Ability to establish connections between different domains of knowledge<br>• Knowledge of the reasons why other projects have failed | • Knowledge of the strengths and weaknesses of the organisation<br>• Ability to anticipate problems that may arise<br>• Knowledge of the techniques for patenting<br>• Ability to plan and organise a research project<br>• Knowledge of the research methodology<br>• Established network of contacts external to the organisation<br>• Established network of people inside the organisation<br>• Decisiveness<br>• Patience<br>• Resiliency |

Part of these competencies coincide with the components that have been considered traditionally inside the concept of intellectual capital, as mentioned by authors like Luthy (1998) and Choong (2008).

Hypothesis H3 is observed in the results, as long as knowledge/tacit knowledge is the most mentioned characteristic of a senior researcher, like it was exposed in the interviews. Knowledge itself, identification of the missing one or the routes to get to it, were some of the attributes identified by the senior researchers about their tacit contributions.

*4.3. Roles of Senior Researchers*

The most frequent role of senior researchers, as mentioned explicitly or implicitly by the interviewees, is mentoring. Each and every one of them refer to this as the one that usually older researchers play, either as counsellors or consultants. Such frequent mention, supports the Hypothesis H4, meaning that mentoring-coaching is perceived as the main role that senior researchers play or could play in their organisations.

"Coaching is very important because you [might] have the knowledge, but you have some intangible things that you can only transfer by a strong day-to-day coaching. And that's also one of the key qualities of a senior researcher: he's to be able to spend time with the others, with the youngest researcher to transfer, to explain them, to help, to support, etc.; and that's something [ . . . ] that should be part of the evaluation for a senior researcher: his/her ability not [only] to develop knowledge, but to let young engineers to grow and let

them develop knowledge in the future, it means: to share what he has learned, to share some way of working, to share ideas, etc.; this is a key for the collective improvement of the research in this company. It is key." (Interview number 5).

In second place, mentioned by two thirds of the interviewees, was the capacity to provide substantial time-savings. Technical knowledge possessed by the veterans, along with the learning-by-doing experience, made them experts in avoiding wrong paths on research that usually will not have fruitful results. It is one additional example of the valuable tacit knowledge they can contribute.

In a more residual way, other aspects were mentioned, such as support in difficult situations and the ability to teach building resilience skills in their younger peers. Landaeta and Kotnour (2008) have already exposed the psychological gains that a mentoring relationship, formal or informal, can bring to a young mentee.

"Almost by definition, part of the job of the senior is that: to share a bit of his experience with those who come to a team so this experience, along with the new skills, new tools and new energy of a young person, is then finally well routed and increased. I would say that it is practically a fundamental part of the work itself" (Interview number 9).

Table 4 shows a compendium of the senior roles that interviewees identified, most times, within the framework of a more or less formal mentoring.

**Table 4.** Roles that a senior researcher could play in an industrial organisation.

| Roles of Senior Researchers in an Industrial Organisation | |
|---|---|
| • Expert<br>• Consultant<br>• Counsellor<br>• Motivator<br>• Organiser | • Coordinator<br>• Facilitator<br>• Empower (of younger people)<br>• Marshal person (who channels people and resources)<br>• Resilience model person |

The potential of mentoring to produce benefits, for both mentors and mentees, has been highlighted by several studies (Hobson 2012). In this research, some senior researchers have been able to recognize that a relationship with young researchers might also bring them great benefits, such as catching up on certain technological developments, or perceiving the enthusiasm and drive shown by juniors. The literature has already identified the self-esteem improvement and empowerment of mentors by the mere recognition as wise men to approach (Dunham and Burt 2011).

Figure 5 represents schematically the results of the characteristics a senior researcher usually has, concerning the roles he can assume, the challenges he usually faces, and the contributions he can propose to a research entity.

As corroborated in H1, the number of patent applications is not a sufficient indicator of productivity, therefore it cannot be used to assess the activity of senior researchers. Moreover, through Sections 4.2 and 4.3 it has been shown that their experience and perspective bring substantial advantages at both team and firm levels along with the inherent technical expertise. Such skills and contextual knowledge cannot be taught by formal education but only can be achieved through experience and development of personal relationships, constituting a clear advantage for seniors compared to junior workers. For those reasons, we feel that Hypothesis 2 (H2) is supported when we stated that seniors´ productivity does not diminish with age but changes instead, as they provide other benefits that cannot be acquired by younger team members. As a result, it was demonstrated that new metrics for research productivity are needed in order to properly reflect the tacit contributions.

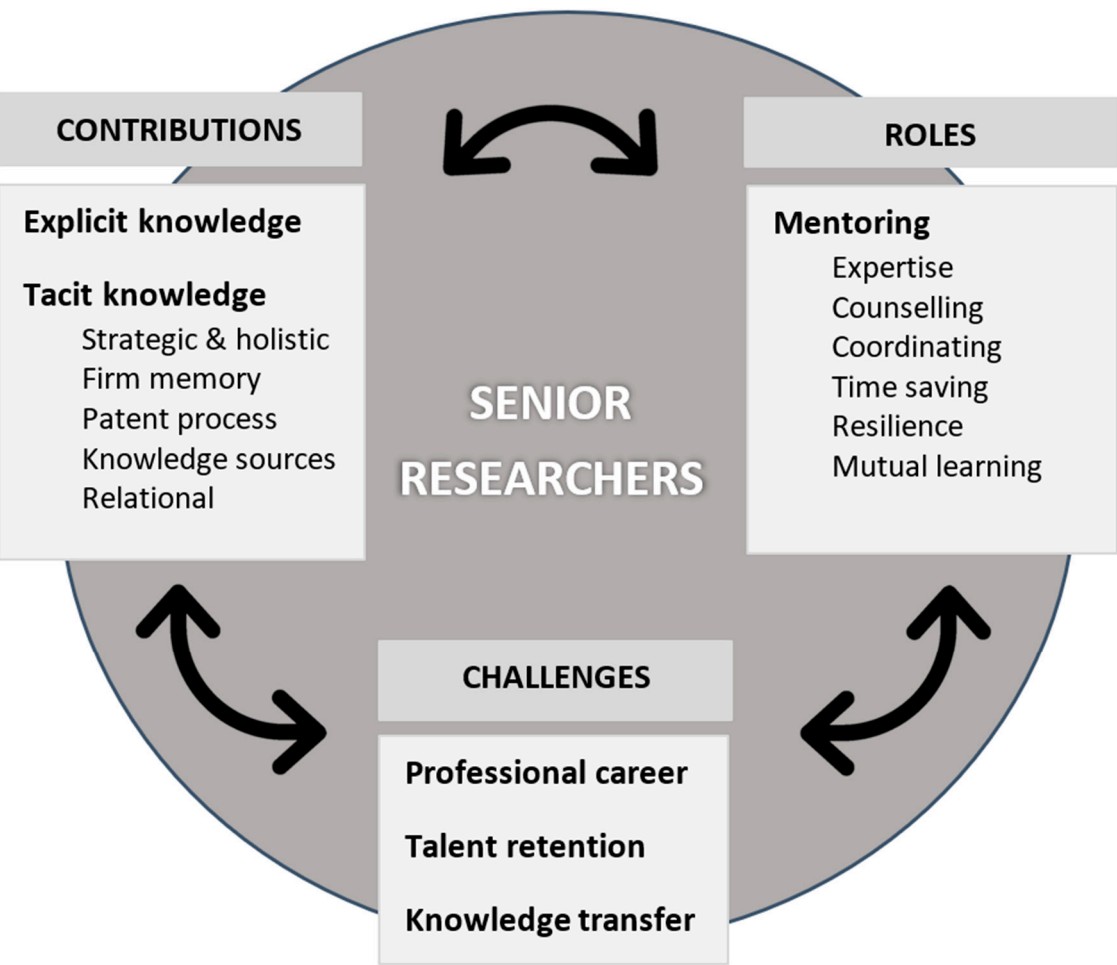

**Figure 5.** Synthesis of results for the study.

### 5. Challenges around Senior Researchers

Interviewees acknowledged that many people drop out of research positions after a certain period of time. There was not a single reason for career/position transformation among researchers, as some of them did not like the area of study anymore or they just wanted to try new domains. A few of them moved to the academic sector, and many stayed in the company in positions such as production, quality, commercial, etc. The studied organisation is a big enterprise, so it is relatively easy to change positions: this circumstances can be positive, since the literature has confirmed that intra-company mobility contributes to tacit knowledge sharing and ideas recombination which, in turn, leads to the improvement of innovation results in distributed organisations (Choudhury 2017).

However, one of the main reasons discussed in the study is the lack of professional career evolution. As mentioned by the interviewees, it is assumed that people who stay in R&D is not going to be promoted in the company, and their prospects are limited. Therefore, the development of a purely research professional career, where some type of recognition, visibility or reward is more formally integrated, seems today to be an evident challenge.

Professional career and talent retention complement each other. Senior researchers, in their mentoring role, can help integrating young talent into the organisation. According to Short (2014), mentoring is not only about adding value to the workforce development strategies, but also helping with a wider range of issues such as retention, engagement, absence prevention and well-being.

Some interviewees highlighted the managerial role that seniors have. As an example, they mentioned the importance of becoming good managers, as well as having the necessary

support from their top management to undertake newer and more novelty research areas or to patent/publish more.

> "A person in R&D has different needs than somebody with a different vocation: having the freedom to investigate and to search, to increase his knowledge and to open new lines of work, are subjects that R&D people value more than those in other sectors." (Interview number 8)

> "Regarding money, after a certain limit where you cover all your needs, other factors enter to fill the job satisfaction, as an example, some work that becomes technically interesting." (Interview number 2)

The last major challenge identified by the interviewees is knowledge transfer: in the studied organisation there are databases with documentation, projects, articles, etc.; several computer tools have been developed, as well as numerous meetings to exchange knowledge. However, there does not seem to be a clear system for the transfer of knowledge linked to experience. Knowledge transfer in this organisation depended largely on the will of the researchers themselves or very specific managers, who usually did not have much time to invest.

Interviewees did not concur on the best system for tacit knowledge transfer: most comments in the interviews related to the work with intergenerational teams and mentoring.

> "Even if it is a difficult duty to be performed by the researchers in the organisation, there should exist a task that consists of generating research management structures in inter-generational teams, so that any [knowledge] transfer is done naturally and not within a month of departure notice [in the case of retirement or job change], in which is impossible to share a whole working life or long years of work" (Interview number 1)

## 6. Conclusions and Future Research

Senior researchers hold extensive knowledge. On the one hand, tangible knowledge, coming from a rather technological nature, seems to be more closely linked to experience in a certain position than to age itself. On the other hand, tacit knowledge, which is more related to the number of years in which the scientist/engineer has carried out his/her role as researcher, and therefore, it is more related to age. Senior workers can become more valuable to an organisation because of their tacit knowledge, instead of their tangible knowledge or their work in the laboratory.

Contributions of senior researchers identified in this study coincided, at least in part, with the aspects found in the literature as part of the intellectual capital of an organisation. As a consequence, it could be stated that seniors contribute intensely to the intellectual capital of the company in which they work.

In the industrial field, traditional indicators of scientific productivity do not fit well. Organisations are not always interested in publishing or patenting and, moreover, not all research topics have the same potential for patentability. In addition, senior researchers use to acquire new roles, in a formal or informal way, related to mentoring and knowledge transfer.

This work provides the vision of senior research workers of a multinational industrial company. Their point of view, not extensively studied to date, is the result of a long and productive research career, and not the one of stereotypes linked to age. Based on this research, those responsible for research teams and human resources have more elements to design strategies and make the most of the senior researchers' contributions.

Finally, as all interviewees belong to the same company, the results obtained in this research might be influenced by their corporate culture and sector's idiosyncrasy, despite its size and multinational nature. Therefore, this study should be consolidated with broader studies, both qualitative and quantitative, incorporating organisations of different sizes and economic sectors.

**Author Contributions:** Conceptualization, A.P.-E. and M.A.V.B.; methodology, A.P.-E.; software, A.P.-E.; validation, W.P.V., A.P.-E. and M.A.V.B.; formal analysis, A.P.-E. and M.A.V.B.; investigation, W.P.V.; resources, W.P.V.; data curation, M.A.V.B.; writing—original draft preparation, W.P.V. and A.P.-E.; writing—review and editing, W.P.V.; visualization, W.P.V.; supervision, A.P-.E.; project administration, M.A.V.B.; funding acquisition, M.A.V.B. All authors have read and agreed to the published version of the manuscript.

**Funding:** This research was funded by the Foundation for the Promotion of Applied Scientific Research and Technology in Asturias, grant number AYUD/2021/50953.

**Institutional Review Board Statement:** Not applicable.

**Informed Consent Statement:** Not applicable.

**Data Availability Statement:** Not applicable.

**Acknowledgments:** The authors wish to thank all respondents of the interviews for allowing us to get to know their position as senior researchers, being noticeable their interest on the subject and the possibility of feedback for improval on their own performance.

**Conflicts of Interest:** W.P.V. certifies that has no conflict of interest but is an employee of the investigated organisation at the moment of the research. A.P.-E. certifies that has no conflict of interest, no affiliations with or involvement in any organisation or entity with any financial or non-financial interest. M.A.V.B. certifies that has no conflict of interest but is a service supplier of the investigated organisation at the moment of the research.

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
