# Peer review of "Tacit Contributions and Roles of Senior Researchers: Experiences of a Multinational Company"

_admsci, doi:10.3390/admsci12040192_

Round 1
Reviewer 1 Report
Interesting research, which may be continued. In future, it would be great to include more respondents, do surveys in different kind of organizations: public, private, etc.
Also, specify the age of senior researchers.You may use other statistical methods as well, for example Pearson Chi square and cross tabulation, to see if there is any significant statistical relationship between some factors.
Author Response
Dear reviewer, thank you for your valuable comments you’ve made to improve our paper. Here below we address your observations and suggestions.
Interesting research, which may be continued. In future, it would be great to include more respondents, do surveys in different kind of organizations: public, private, etc.
Answer: Thanks, we’ve indeed continue working on this research. At least for this manuscript and as mentioned in the document: most of the interviewee answers were very uniform, causing saturation, forcing the authors to limit the number of interviewees to 10, following the considerations of Baker and Edwards (2012). We added one more person giving her point of view as the one responsible for patenting in the organization. For future research, we’d considered performing quantitative studies in different kind of organizations, as you suggested.
Also, specify the age of senior researchers.
Answer: We mentioned the age of the senior researchers as age ranges in the table 1. Additionally, we added the definition of senior researcher, from our point of view, at the end of the first paragraph of the section 3 (Materials and methods): “It has been determined by the authors that a senior researcher is a person with a technical background (engineering or sciences studies) and more than 15 years of experience in research. To enrich the answers of the study and avoid the saturation of feedback, the authors invited also some researchers, close to the proposed age range, that were well compromised on research, as confirmed by other interviewees.”
You may use other statistical methods as well, for example Pearson Chi square and cross tabulation, to see if there is any significant statistical relationship between some factors.
Answer: We consider this suggestion for future stages of this research, which will be quantitative, and the use of these relationships will enrich it.
Reviewer 2 Report
Firstly I would like to congratulate to authors very interesting paper covering topic of senior researchers in multinational company. It's worth to mention this topic could be valuable for scientific researchers in management as well for management practitioners.
Introduction, literature review and materials and methods section seems to be prepared very well (some issues raises due to further results presentations). Literature items cited in the paper are important for its content.
Some problems raises with research question and hypotheses formulation. Firstly, main research question should be single. Secondly, what are relations between research question and hypotheses? Such discussion should be conducted. To be honest, if authors are using qualitative (inductive) research method it may mean that we should operate on main research question and detailed ones (the structure of study is clear for reader in such situation). In the text I couldn't find clear statement about status of H3 and H4. Making a revision of this issues could turn into better presentation/structure of results and discussion section content.
I recommend to inform readers about coding reliability index. NVivo offers a set of measures regrading this issue. It could provide crucial information about coding compatibility in the study. Tables 3, 4 present information in two columns and with different layer (why?). Please explain who is senior researcher in your study. Is it position in the company? Is it a person with sufficient professional experience? Is it maybe a person in some age?
Good luck with revisions and publication of your paper.
Author Response
Dear reviewer, thank you for your valuable comments you’ve made to improve our paper. Here below we address your observations and suggestions.
Firstly I would like to congratulate to authors very interesting paper covering topic of senior researchers in multinational company. It's worth to mention this topic could be valuable for scientific researchers in management as well for management practitioners.
Introduction, literature review and materials and methods section seems to be prepared very well (some issues raises due to further results presentations). Literature items cited in the paper are important for its content.
Answer: Thanks, we continue working on this research and we hope our contribution will facilitate help on the management of experienced research teams. For the raised issues, we hope our modifications facilitate the understanding on the results.
Some problems raises with research question and hypotheses formulation. Firstly, main research question should be single.
Answer: We propose the research question in the fourth paragraph of the first section (Introduction), but as proposed, we have simplified the question to a single one: “RQ: What are the tacit contributions that older researchers provide within an organisation, contributing to its inventive productivity, to a greater or lesser extent?
Secondly, what are relations between research question and hypotheses? Such discussion should be conducted. To be honest, if authors are using qualitative (inductive) research method it may mean that we should operate on main research question and detailed ones (the structure of study is clear for reader in such situation).
Answer: The research question is the basis of the study, which has been done adopting the qualitative research technique as a deductive one. The hypotheses of the study are the result of the bibliographical analysis and the biographical research done with the interviewees, performed using a scholarly chronicle style (Eaton, 1964), which focuses on the historical path of the person, telling his/her story in chronological order with emphasis upon developments of particular plots on his/her scientific development, including detailed description of particular acts of recognition or notoriety. The hypotheses born in the bibliographical, that initially was expected to reply somehow the research question, have guided the design of the questions performed to the interviewees and allow us to obtain priceless contributions from them.
In the text I couldn't find clear statement about status of H3 and H4. Making a revision of this issues could turn into better presentation/structure of results and discussion section content.
Answer: We agree with you, both hypotheses are not explicitly exposed in the conclusions of the paper. For H3, we do consider tacit knowledge itself is one of the most important contributions of senior researchers to its organization. For H4, mentoring-coaching hypothesis was mentioned extensively by the interviewees.
We have included at the end of the section 4.2 (Results and discussion / Tacit contributions of senior researchers) the following: “Hypothesis H3 is observed in the results, as long as knowledge/tacit knowledge is the most mentioned characteristic of a senior researcher, as exposed in the interviews. Knowledge itself, identification of the missing one or the routes to get to it, were some of the attributes identified by the senior researchers about their tacit contributions”.
It has been added at the end of the first paragraph of the section 4.3 (Results and discussion / Roles of senior researchers) the following: “Such frequent mention, supports the Hypothesis H4, meaning that mentoring-coaching is perceived as the main role that senior researchers play or could play in their organisations.”
I recommend to inform readers about coding reliability index. NVivo offers a set of measures regrading this issue. It could provide crucial information about coding compatibility in the study.
Answer: The coding of the interviews was done by one of the authors, with the consensus of the three. Three rounds of relational reading of the transcription and codification were done and, as mentioned in the manuscript, “The authors discussed the result of this new codification until they reached a consensus”.
However, to be clearer on the coding, we’ve removed the previous line and included on the last paragraph of the section 3 (Materials and methods), the following: “All three rounds of coding were performed by one of the authors, but each one was reached by the consensus of the three researchers of the study”.
Tables 3, 4 present information in two columns and with different layer (why?).
Answer: It was an edition mistake. We have removed the bold font, justified align to the left and changed the hyphen to bullets, to easy its reading.
Please explain who is senior researcher in your study. Is it position in the company? Is it a person with sufficient professional experience? Is it maybe a person in some age?
Answer: we added the definition of senior researcher, from our point of view, at the end of the first paragraph of the section 3 (Materials and methods): “It has been determined by the authors that a senior researcher is a person with a technical background (engineering or sciences studies) and more than 15 years of experience in research. To enrich the answers of the study and avoid the saturation of feedback, the authors invited also some researchers, close to the proposed age range, that were well compromised on research, as confirmed by other interviewees.”